# The Behaviours in Dementia Toolkit: A Descriptive Study on the Reach and Early Impact of a Digital Health Resource Library About Dementia-Related Mood and Behaviour Changes

**DOI:** 10.3390/geriatrics10030079

**Published:** 2025-06-11

**Authors:** Lauren Albrecht, Nick Ubels, Brenda Martinussen, Gary Naglie, Mark Rapoport, Stacey Hatch, Dallas Seitz, Claire Checkland, David Conn

**Affiliations:** 1Canadian Coalition for Seniors’ Mental Health, Markham, ON L3R 9X9, Canada; lauren.albrecht@gmail.com (L.A.); martinub@yahoo.ca (B.M.); dallas.seitz@ucalgary.ca (D.S.); ccheckland@ccsmh.ca (C.C.); 2Department of Medicine and Institute of Health Policy, Management and Evaluation, University of Toronto, Toronto, ON M6A 2E1, Canada; gnaglie@baycrest.org; 3Department of Medicine and Rotman Research Institute, Baycrest, Toronto, ON M6A 2E1, Canada; 4Department of Psychiatry, University of Toronto, Toronto, ON M5T 1R8, Canada; mark.rapoport@sunnybrook.ca; 5Sunnybrook Health Sciences Centre, Toronto, ON M5T 1R8, Canada; 6Faculty of Behavioural Sciences, Yorkville University, Fredericton, NB E3C 2R9, Canada; 18seh@queensu.ca; 7Cummings School of Medicine, Department of Psychiatry, University of Calgary, Calgary, AB T2N 1N4, Canada; 8Department of Psychiatry, Baycrest, Toronto, ON M6A 2E1, Canada

**Keywords:** behaviours in dementia, behavioural and psychological symptoms of dementia, responsive behaviours, internet, mHealth, mobile health, digital health, eHealth, library, digital library, health information resources, caregiving, carer, formal caregiver, informal caregiver, care partner

## Abstract

**Background:** Dementia is a syndrome with a high global prevalence that includes a number of progressive diseases of the brain affecting various cognitive domains such as memory and thinking and the performance of daily activities. It manifests as symptoms which often include significant mood and behaviour changes that are highly varied. Changed moods and behaviours due to dementia may reflect distress and may be stressful for both the person living with dementia and their informal and formal carers. To provide dementia care support specific to mood and behaviour changes, the Behaviours in Dementia Toolkit website (BiDT) was developed using human-centred design principles. The BiDT houses a user-friendly, digital library of over 300 free, practical, and evidence-informed resources to help all care partners better understand and compassionately respond to behaviours in dementia so they can support people with dementia to live well. **Objective:** (1) To characterize the users that visited the BiDT; and (2) to understand the platform’s early impact on these users. **Methods:** A multi-method, descriptive study was conducted in the early post-website launch period. Outcomes and measures examined included the following: (1) reach: unique visitors, region, unique visits, return visits, bounce rate; (2) engagement: engaged users, engaged sessions, session duration, pages viewed, engagement rate per webpage, search terms, resources accessed; (3) knowledge change; (4) behaviour change; and (5) website impact: relevance, feasibility, intention to use, improving access and use of dementia guidance, recommend to others. Data was collected using Google Analytics and an electronic survey of website users. **Results:** From 4 February to 31 March 2024, there were 76,890 unique visitors to the BiDT from 109 countries. Of 76,890 unique visitors to the BiDT during this period, 16,626 were engaged users as defined by Google Analytics (22%) from 80 countries. The highest number of unique engaged users were from Canada (*n* = 8124) with an engagement rate of 38%. From 5 March 2024 to 31 March 2024, 100 electronic surveys were completed by website users and included in the analysis. Website users indicated that the BiDT validated or increased their dementia care knowledge, beliefs, and activities (82%) and they reported that the website validated their current care approaches or increased their ability to provide care (78%). Further, 77% of respondents indicated that they intend to continue using the BiDT and 81.6% said that they would recommend it to others to review and adopt. **Conclusions:** The BiDT is a promising tool for sharing practical and evidence-informed information resources to support people experiencing dementia-related mood and behaviour changes. Early evaluation of the website has demonstrated significant reach and engagement with users in Canada and internationally. Survey data also demonstrated high ratings of website relevance, feasibility, intention to use, knowledge change, practice support, and its contribution to dementia guidance.

## 1. Introduction

Dementia is a syndrome that includes a number of progressive diseases of the brain affecting memory, thinking and the performance of daily activities [1]. Globally, more than 55 million people have dementia; this figure is projected to triple by 2050 [2].

Dementia can manifest as symptoms across a number of domains [3]. Dementia-related mood and behaviour changes, often called the behavioural and psychological symptoms of dementia or responsive behaviours, affects nearly all people living with dementia at some point during their disease course [4]; however, the ways in which these changes are individually expressed varies widely [5]. Changed moods and behaviours due to dementia are highly complex, varied, and stressful [4,5,6], and may reflect distress and unmet needs from the person living with dementia [7]. These changes are associated with deterioration of relationships, increased caregiver burden, institutionalization, and mortality [8]. A research study identified that an increase in caregiver burden was found to increase the frequency and severity of the behaviours [9].

In the USA, approximately 65% of people with dementia live at home [10] and 60% of people in long term care have dementia [10,11]. In low- and middle-income countries, as many as 94% of people living with dementia reside at home. A longitudinal study determined that 97% of people living with dementia in community settings experienced at least one behaviour or mood symptom, most commonly depression, apathy, or agitation [12]. The occurrence of dementia-related mood or behaviour changes are associated with increased long-term care (nursing home) placement for people living with dementia [13]. This means that formal (i.e., professional) and informal (i.e., non-professional) caregivers including spouses, family members, home care staff, care aides, nursing staff, allied health professionals, and physicians will be involved in their care. Formal and informal care partners need assistance to cope with dementia-related mood and behaviour changes to help ease the emotional and physical strain of caregiving [14] and reduce the distress experienced by the person living with dementia [9].

Health information websites are a common source of complex health information [15]. The internet is the most frequently used venue for older adults seeking health information; however, it is simultaneously the least trusted [16]. Online health information should mitigate this mistrust by demonstrating credibility, supporting ease of use, and reducing information overload [16]. Health care providers are increasingly required to use information and communication technologies in their work; however, uptake has not been optimal [17]. Due to the ubiquitous use of the internet and the different ways varied information seekers obtain appropriate information and use it in their lives, health information websites are challenging to design for high usability and maximum benefit [15].

To support the physical, mental, and social wellbeing of people living with dementia, the Canadian Coalition for Seniors’ Mental Health (CCSMH) developed the Behaviours in Dementia Toolkit (BiDT). The BiDT is a free, English-language website (https://behavioursindementia.ca/ (accessed on 3 May 2024)) that contains a user-friendly, digital library of over 300 free resources about dementia-related mood and behaviour changes for informal and formal care partners [18]. The goal of the BiDT is to help all care partners, both formal and informal, better understand and compassionately respond to behaviours in dementia so they can support people with dementia to live well. See Appendix A for an overview of the website structure and content.

A human-centred design approach was used to develop the BiDT, which considers the multiplicity of users and systems in the design of an innovation and is widely regarded as an effective approach for creating digital health innovations [19,20]. Employing a human-centred design approach in medical devices has also been shown to reduce the need for modifications and updates after they have been made publicly available, supporting cost-effectiveness [21]. A 2020 systematic review indicated that co-designed technologies positively impacted disease knowledge, access to health care, patient satisfaction, health outcomes, medical errors, and medical costs [22]. It is a particularly salient approach for developing digital health solutions for older adults [21,23] and specifically for developing effective health websites [15]. Appendix A provides a detailed overview of the human-centred design phases and activities that took place in the development process for the BiDT. Three separate publications detail various aspects of the BiDT development and design process [16,24,25].

The BiDT project team conducted a number of activities to create a person-centred and user-friendly platform that would attend to diverse audience needs. The objective of this study was to characterize the users that visited the BiDT in the early launch period and understand the platform’s early impact on these users.

## 2. Materials and Methods

The BiDT was launched to the public in February 2024. Promotion of the website was comprehensive and extensive from January 2024 through March 2024. Outreach activities targeted to audiences in Canada included direct emails and presentations; a press release and media engagement; sponsored articles in digital and print magazines; and distribution of 26,000 postcards publicizing the BiDT. International outreach activities included social media posts and Google, Facebook, and YouTube advertisements. Collectively, these efforts connected us with a potential audience of approximately seven million people.

The subject of this study was to determine whether the development and implementation of the BiDT was successful. We conducted a multi-method, descriptive study in the early post-launch period (i.e., up to two months post-launch) to address the following research questions:How many users visited the BiDT?How did users engage with the BiDT?How did users rate their change in knowledge about behaviours in dementia and effective responses after using the BiDT?How did users anticipate their care may change after using the BiDT?How did users rate the contribution of the BiDT to dementia guidance in Canada?

These research questions were mapped to five main outcomes and nineteen outcome measures (Table 1). Our study received research ethics board approval from Baycrest (REB# 24-07).

### 2.1. Google Analytics

Reach and engagement outcome measures were collected via Google Analytics (https://marketingplatform.google.com/about/analytics/ (accessed 5 November 2023)) [26]. Unique visitors are distinct, unduplicated users defined by IP address; traffic from bots and spiders is automatically excluded [27,28]. Google Analytics 4 defines engagement metrics as follows: (1) engaged users have completed a session lasting longer than 10 s, having a key event, or having at least two page or screen views; (2) engaged sessions last longer than 10 s, have a key event, or have at least two page views or screen views; (3) bounce rate is the percentage of sessions that were not engaged; and (4) engagement rate measures the percentage of visitors who interacted with content or spent a significant amount of time on a specific page [29].

Data were collected from 4 February 2024 to 31 March 2024. Google Analytics cookies collect information including IP address, device type and operating system, referring URLs, geolocation, and pages visited. This information is de-personalized and not connected to any individual. Website users were advised of our use of Google Analytics with our website privacy policy and provided tools and instructions to opt out [30].

### 2.2. Electronic Survey

Knowledge change, behaviour change, and website impact measures were collected through a voluntary electronic survey of website users via Qualtrics (https://www.qualtrics.com/ (accessed 29 October 2023)) (Appendix A) [31]. These questions were required by the project funder. The survey and survey recruitment were aimed at Canadian website users and available in English-only due to time and funding constraints. Survey recruitment took place from 5 March to 31 March 2024. Website users were invited to participate in the survey through (1) a banner with an invitation and survey link in the website header; (2) two Facebook advertisements; (3) e-newsletter and social media posts; and (4) direct emails from members of the Behaviours in Dementia Working Group to their networks. Potential participants were first presented with a digital information letter about the study at the survey link. Advancing to the survey implied their consent to participate. Once the data collection period ended, the survey was closed for data collection and the Excel file was downloaded from Qualtrics to CCSMH’s secure server for cleaning, analysis, and secure, long-term storage. Survey responses were included in the analysis if all demographic items and at least one outcome measure were completed.

### 2.3. Data Analysis

Quantitative data from Google Analytics and the electronic survey were analyzed using descriptive statistics. Google Analytics 4 [32] and Microsoft Excel (https://www.microsoft.com/en-ca/microsoft-365/excel (accessed on 4 June 2023)) [33] were used for analysis.

## 3. Results

### 3.1. Google Analytics

From 4 February to 31 March 2024, there were 76,890 unique visitors to the BiDT from 109 countries. The highest number of unique visitors were from the Philippines (*n* = 31,816); Canada (*n* = 21,530); Nigeria (*n* = 8086); Malaysia (*n* = 3962); and Uganda (*n* = 2319). Within Canada, where the BiDT was developed, unique visitors opened the BiDT website from all provinces and territories.

Of 76,890 unique visitors to the BiDT during this period, 16,626 were engaged users (22%) from 80 countries. The highest number of unique engaged users were from Canada (*n* = 8124) with an engagement rate of 38%. Unique engaged users in Canada visited the BiDT from all provinces and territories.

Reach and engagement outcome measures and results are described in Table 2. The results are reported for two groups: total visitors who came to the website and the sub-group of visitors with engaged sessions on the website.

The most often used keyword terms used to search the library catalogue were (1) driving (*n* = 22); (2) won’t eat (*n* = 6); and (3) apathy (*n* = 5). Of the 318 resources in the collection, the five most frequently viewed were (1) caregiver tip sheet: anger, frustration, and fidgeting (*n* = 604); (2) supporting independence and personhood (*n* = 367); (3) five seriously fun ideas from a recreation therapist (*n* = 278); (4) effective online supports help navigate behaviours in dementia (*n* = 245); and (5) ten helpful resources to increase meaningful engagement (*n* = 169).

### 3.2. Electronic Survey

From 5 March 2024 to 31 March 2024, 297 people accessed the survey information page; 230 people consented to participate (77% consent rate); 104 started the survey (45% response rate); and 100 surveys were included in the analysis (96% completion rate).

All survey respondents resided in Canada in the following regions: Ontario (*n* = 49, 49%); British Columbia (*n* = 18, 18%); Alberta (*n* = 11, 11%); Nova Scotia (*n* = 6, 6%); Manitoba (*n* = 6, 6%); Saskatchewan (*n* = 5, 5%); New Brunswick (*n* = 2, 2%); Newfoundland and Labrador (*n* = 1, 1%); Northwest Territories (*n* = 1, 1%); and Prince Edward Island (*n* = 1, 1%). There were no respondents from Nunavut, Quebec, or Yukon.

Survey respondents identified as (multi-response) caring for someone living with dementia (*n* = 40, 40%); having a general interest in dementia (*n* = 33, 33%); knowing someone living with dementia (*n* = 27, 27%); working in a clinical or non-clinical role connected to dementia (*n* = 17, 17%); other role (*n* = 17, 17%); or someone living with dementia (*n* = 3, 3%).

Knowledge change, behaviour change, and website impact results are reported in Table 3.

## 4. Discussion

### 4.1. Reach and Engagement

The Google Analytics results showed high traffic from users in the Philippines, Canada, Nigeria, Malaysia, and Uganda. This is likely a result of our Google, Facebook, and YouTube advertisements targeting English-speaking countries around the world, which were optimized towards low-middle income countries where the cost per click is less (e.g., cost for users in Philippines is PHP 0.02 per click vs. the United States at USD 1.21 per click). It may also reflect less access to health information and care in Sub-Saharan Africa and Southern Asia [34]. The second largest group and most engaged users were from Canada. This was expected as the project took place in Canada and incorporated extensive outreach activities across Canada.

The bounce rate of 71% was high compared to the average bounce rate for health care websites of 58% [35]. Users tend to form an opinion about a website in 0.05 s [36], so it is likely that our paid advertisements through Google and Facebook were not optimally targeted to our intended audiences, which is likely why they did not remain on the website beyond ten seconds or engage with web content. We would expect the bounce rate to improve as our more targeted and sustained outreach activities (e.g., postcard distribution; direct emails and meetings; library listings, etc.) grow our engaged user base and the BiDT becomes more deeply embedded in the dementia care information ecosystem. This data point should be examined at regular intervals over time to ensure that our strategies are working as intended to attract users who find the website useful to support their caregiving.

For the sub-set of engaged users of the BiDT, our results demonstrate a high number of users and sessions. During the study period, 16,626 engaged users visited the website; a figure nearly seven times higher than the number of average users for health care websites [34]. Additionally, those users had 23,714 sessions on the website, which is nearly six times higher than the average number of sessions for health care websites [35]. This is likely a result of the novelty of the website and points to the BiDT meeting a real and present need for informal and formal care partners of people experiencing mood issues and behaviours in dementia. The usefulness of the website is bolstered by high engagement rates on the additional main navigation (i.e., level 1) pages in the BiDT (i.e., other than the home page). For comparison, the all industry average engagement rate for level 1 pages is 75% and the average engagement rate of level 1 pages for non-profit organizations is 53% [37]. Level 1 page engagement rates ranging from 84% to 94% confirm that the BiDT is relevant and engaging; readable and scannable; and facilitates a positive user experience [37].

The return views for both the overall group (21,204, 28%) and engaged users’ sub-group (7088, 43%) demonstrates that the BiDT met its value proposition for providing practical and evidence-informed resources to support people experiencing behaviours in dementia. Users returned, even in the short study timeline, which means that they found something of value on the website or in the library. This data point should also be tracked over time to ensure the BiDT remains relevant to our key audiences.

### 4.2. Knowledge Change, Behaviour Change, and Website Impact

The results of the electronic survey further highlighted that the BiDT may help carers of people living with dementia. More than 80% of website users indicated that the BiDT validated (40%) or increased (42%) their dementia care knowledge, beliefs, and activities and more than 75% of website users indicated that the BiDT validated their current care approaches (44%) or increased their ability to provide care (34%). Further, over three quarters of respondents (78%) indicated that they intend to continue to use the BiDT and over four fifths (86%) said that they would recommend it to others to use. Additional high ratings on relevance (76%), feasibility (75%), and improved access to dementia guidance (80%) demonstrate that the BiDT will be a highly accessed tool into the future. Particularly as the BiDT becomes more widely known, the use of the website becomes integrated into practice, and users return for more and different resources to meet their needs.

### 4.3. Limitations

The BiDT is currently an English language website, though it does contain 122 multilingual resources in 12 different languages. The evaluation survey was also only conducted in English, which limits its generalizability, and the data presented is a snapshot immediately post-launch when we had considerable capacity for outreach. Ongoing monitoring of reach and engagement metrics will confirm whether the BiDT remains relevant to users over time. Expansion of the BiDT to other languages would also facilitate greater uptake in Canada, which is an officially bilingual English/French country with over 200 other languages represented in the population [38], and worldwide.

For the electronic survey, fewer participants filled out the survey after consenting (45%). This attrition could be attributed to the request for participants to visit the BiDT before initiating the survey if they had not yet done so and providing an external link. While the drop in participation was not ideal, it was important that users provide an honest assessment of their website experience, which required them to have used the BiDT. Of those that returned or continued with the survey, we observed an excellent survey completion rate (96%).

To understand the sustained impact of the website, it will be important to conduct a future similar survey to solicit feedback from a higher proportion of formal care partners, from participants proportionally representing all provinces and territories, and from non-English speaking users to understand the perspectives, needs, and contextual nuances of different roles and locations. The addition of qualitative data collection (e.g., survey questions, focus groups, or interviews) may also be important to solicit users’ positive and negative feedback in a more open-ended and detailed fashion, as well as to explore more complex considerations including digital literacy.

## 5. Conclusions

Previous research has demonstrated the critical importance of thoughtful design for digital interfaces to support ease of use, particularly for older adults, including consistent interface and terminology and clear processes [20,39], leading to a user-friendly experience [40,41]. The objective of this study was to characterize the users that visited the BiDT in the early launch period and understand the platform’s early impact on these users. Our findings demonstrated that employing a human-centred design approach in the development of the BiDT led to high reach and engagement as well as positive knowledge change, behaviour change, and website impact results. Longer term evaluation is needed to monitor these metrics over time to ensure sustained enthusiasm for and usefulness of the website. Future investments in the website could also help expand its offerings to reflect emerging evidence and the unique contexts of a variety of equity deserving groups. Sustained outreach and engagement activities will facilitate the integration of this useful tool in dementia care information ecosystems.

## Figures and Tables

**Table 1 geriatrics-10-00079-t001:** Study questions, outcomes, and measures.

Research Question	Outcome	Outcome Measure	Data Collection Tool
How many users visited the website?	Reach	number of unique visitors number of unique visitors by region number of unique visits number of return visits bounce rate (definition below)	Google Analytics
How did users engage with the website?	Engagement	engaged users (definition below) engaged sessions (definition below) average session duration average pages viewed per session engagement rate per webpage (definition below) top keyword search terms used top resources accessed	Google Analytics
How did users rate their change in knowledge about behaviours in dementia and effective responses?	Knowledge change	change in knowledge after using the BiDT	Electronic survey
How did users anticipate their care may change?	Behaviour change	change in ability to provide care to people experiencing behaviours in dementia after using the BiDT	Electronic survey
How did users rate the website’s contribution to dementia guidance in Canada?	Website impact	relevance of the BiDT feasibility of the BiDT intention to continue to use the BiDT the BiDT makes a meaningful contribution to improving access and use of dementia guidance and support in Canada would recommend the BiDT to others to use	Electronic survey

**Table 2 geriatrics-10-00079-t002:** Reach and engagement outcome measures and results.

Outcome Measures	Results
**Total visitors**	
Total number of unique visitors	76,890
Number of unique views	98,094
Bounce rate	70.9%
Number of return views	21,204
Mean session duration, minutes/seconds	1:49
Mean pages per session	1.64
Engagement rate for homepage	28.9%
Engagement rate for behaviours in dementia information page	84.2%
Engagement rate for search library page	93.7%
Engagement rate for advanced search page	91.3%
Engagement rate for recommended resources page	90.0%
Engagement rate for about page	86.9%
**Engaged users**	
Number of unique engaged users	16,626
Number of unique views	23,714
Number of return views	7088
Mean session duration, minutes/seconds	3:26
Mean pages per session	2.65
Engagement rate of users in Philippines	15.0%
Engagement rate of users in Canada	37.7%
Engagement rate of users in Nigeria	14.5%
Engagement rate of users in Malaysia	16.5%
Engagement rate of users in Uganda	16.9%

**Table 3 geriatrics-10-00079-t003:** Knowledge change, behaviour change, and website impact outcome measures results.

Outcome	Survey Question (Number and Percentage of Respondents Completing the Question)	Response (Number, Percentage of Respondents)
Change in knowledge	How did the BiDT impact your knowledge about behaviours in dementia (*n* = 100, 100.0%)?	I had no change in knowledge (*n* = 18, 18.0%) The BiDT validated my current knowledge, beliefs, and activities (*n* = 40, 40.0%) The BiDT increased my knowledge of the topic (*n* = 42, 42.0%)
Change in behaviour	How did the BiDT impact your ability to provide care to people experiencing behaviours in dementia (*n* = 97, 97.0%)?	I had no change in my ability to provide care (*n* = 21, 21.6%) The BiDT validated my current approaches to care (*n* = 43, 44.3%) The BiDT increased my ability to provide care (*n* = 33, 34.0%)
Website impact	The BiDT is relevant to your situation (*n* = 84, 84.0%)	Strongly Disagree (*n* = 2, 2.4%) Somewhat Disagree (*n* = 3, 3.6%) Neither Agree or Disagree (*n* = 15, 17.9%) Somewhat Agree (*n* = 30, 35.7%) Strongly Agree (*n* = 34, 40.5%)
	The BiDT is feasible to use in your situation (*n* = 83, 83.0%)	Strongly Disagree (*n* = 2, 2.4%) Somewhat Disagree (*n* = 4, 4.8%) Neither Agree or Disagree (*n* = 15, 18.1%) Somewhat Agree (*n* = 27, 32.5%) Strongly Agree (*n* = 35, 42.1%)
	The BiDT is something you intend to use (*n* = 86, 86.0%)	Strongly Disagree (*n* = 3, 3.5%) Somewhat Disagree (*n* = 3, 3.5%) Neither Agree or Disagree (*n* = 13, 15.1%) Somewhat Agree (*n* = 30, 34.9%) Strongly Agree (*n* = 37, 43.0%)
	The BiDT makes a meaningful contribution to improving access and use of dementia guidance in Canada (*n* = 83, 83.0%)	Strongly Disagree (*n* = 3, 3.6%) Somewhat Disagree (*n* = 2, 2.4%) Neither Agree or Disagree (*n* = 12, 14.5%) Somewhat Agree (*n* = 21, 25.3%) Strongly Agree (*n* = 45, 54.2%)
	I would recommend the BiDT to others to use (*n* = 83, 83.0%)	Strongly Disagree (*n* = 2, 2.4%) Somewhat Disagree (*n* = 3, 3.6%) Neither Agree or Disagree (*n* = 7, 8.4%) Somewhat Agree (*n* = 22, 26.5%) Strongly Agree (*n* = 49, 59.0%)

## Data Availability

The raw data supporting the conclusions of this article will be made available by the authors on request.

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
