# Peer review of "The Behaviours in Dementia Toolkit: A Descriptive Study on the Reach and Early Impact of a Digital Health Resource Library About Dementia-Related Mood and Behaviour Changes"

_geriatrics, 2025, doi:10.3390/geriatrics10030079_

Round 1

Reviewer 1 Report

Comments and Suggestions for Authors

What was not exactly clear - was the survey accessible only out of Canada?

Author Response

Comment 1: What was not exactly clear - was the survey accessible only out of Canada?

Response 1: While the website was available globally, the survey was only for Canadian-users. Content has been edited in lines 203 to specify Canadian recruitment for the survey. Please see lines 255-259 for a breakdown of survey responses by province/territory. Methods and Conclusions sections have also been edited to enhance clarity.

We appreciate your feedback!

Reviewer 2 Report

Comments and Suggestions for Authors

This manuscript presents the Behaviours in Dementia Toolkit (BiDT), a website to support caregivers of subjects with dementia, addressing their mood and behavioral changes. In the relatively short study period (almost 2 months), over 75 thousand visitors were observed, which is a considerable number. Still, the survey participation was low.

The study has received relevant ethics approval, describes consent procedures, and discloses funding and potential conflicts of interest.

There are, though, a few major concerns which make it impossible to formulate a detailed review:

  • Limited qualitative insight: The study mixes qualitative and quantitative approaches. The manuscript lacks the qualitative results – their presentation and analysis.
  • Quantitative issues: As for the quantitative part, the underlying survey is not presented and discussed in its entirety.

Minor comments.

  • The Introduction does not provide the goal of the study. Restructure.
  • The manuscript is generally well-organised and clearly written. Some sections (e.g., tables) may benefit from minor formatting improvements for legibility.
  • Survey participants were self-selected and thus may have been more positively disposed toward the toolkit. The 45% response rate after consent suggests a potential attrition bias that should be discussed.

Author Response

Thank you for your feedback - please see below for our itemized responses.

Comment 1 - Limited qualitative insight: The study mixes qualitative and quantitative approaches. The manuscript lacks the qualitative results – their presentation and analysis.

Response 1: No qualitative data were collected from Google Analytics. The electronic survey had 1 qualitative question (#3), which is not presented as part of this study due to scope. Please see lines 357-360 for our discussion of qualitative data collection as important for future work. We do have 2 other manuscripts (1 published, 1 in peer-review) discussing different aspects of the website development process, which contain qualitative data collection and analysis on various aspects not related to this paper. Please see supplementary file 1, appendix B Table S4 #s 21, 24 & 26 for details of these other publications.

Comment 2 - Quantitative issues: As for the quantitative part, the underlying survey is not presented and discussed in its entirety.

Response 2: The full set of survey questions are presented in supplementary file 1, Appendix C.

Minor comments.

Comment 3: The Introduction does not provide the goal of the study. Restructure.

Response 3: Please see lines 124-127 for revised statement on study purpose.

Comment 4: The manuscript is generally well-organised and clearly written. Some sections (e.g., tables) may benefit from minor formatting improvements for legibility.

Response 4: Please see Table 1, Table 2 & Table 3 for formatting changes (e.g., font style, font size, table spacing, table sizing) to improve legibility.

Comment 5: Survey participants were self-selected and thus may have been more positively disposed toward the toolkit. The 45% response rate after consent suggests a potential attrition bias that should be discussed.

Response 5: Please see lines 338-352 for a discussion of the response rate limitations. After consenting to the survey, participants were asked to confirm they had viewed the website and redirected to the website if they had not. We believe the drop in response rate is due to users leaving to view the website because they had not already done so and ultimately not returning to the survey. Users that continued with the survey past this point had a very high completion rate (96%).

Reviewer 3 Report

Comments and Suggestions for Authors

This study presents a descriptive evaluation of the Behaviours in Dementia Toolkit (BiDT), a digital health resource designed to support formal and informal caregivers in managing dementia-related mood and behaviour changes. Using a multi-method approach—including Google Analytics and a user survey—the authors assessed the website’s early reach, user engagement, and perceived impact. The findings indicate that BiDT had a significant international reach shortly after launch and was well-received by users, with most reporting improved knowledge, confidence in caregiving, and intent to recommend the resource.

Lines 64–91: Streamline the background to reduce redundancy/ focus more on the gap the BiDT intends to fill.

Lines 214–249: Consider restructuring the Discussion to separate implications from reflections on method.

Line 289: Conclusions should directly tie back to the study questions listed in line 130.

Line 42: Clarify how "unique visitors" were defined and whether bots were excluded.

Lines 198–204: Discuss the lack of survey representation from major regions 

Lines 266–267: Consider stating how the toolkit is expected to remain sustainable and up-to-date.

Lines 268–288: Expand on how language barriers or digital literacy may have limited broader engagement.

Lines 135–137: Add clarity about how "impact" is measured — the current phrasing blurs subjective feedback with measurable outcomes.

Lines 171–173: Clarify how Excel was used for data analysis — were any statistical tests attempted?

Lines 256–258: Reword to reflect that the toolkit "may" support care improvements, as self-reported impact does not confirm behavior change.

Author Response

We appreciate your feedback - please see below for our itemized responses.

Comment 1 - Lines 64–91: Streamline the background to reduce redundancy/ focus more on the gap the BiDT intends to fill.

Response 1: Please see content edits for added description about the gap the BiDT intends to fill in lines 111-125. Also see Supplementary file, Appendix A for more specific information on the content and topics and intended audience for various website sections, content, and resources.

Comment 2 Lines 214–249: Consider restructuring the Discussion to separate implications from reflections on method.

Response 2: Thank you for your feedback. We have reviewed the Discussion and feel it best presents our synthesis.

Comment 3- Line 289: Conclusions should directly tie back to the study questions listed in line 130.

Response 3: Content was edited to include the objective statement and link to our findings in lines 365-367.

Comment 4 - Line 42: Clarify how "unique visitors" were defined and whether bots were excluded.

Response 4: A definition for unique visitors and a short statement is also added about the removal of bots was added to Material & Methods section 2.1 Google Analytics line 186-187.

Comment 5 - Lines 198–204: Discuss the lack of survey representation from major regions 

Response 5: Our main goal was to understand Canadian users of the BiDT – see edits in line 203. Survey respondents were from all provinces/territories except Nunavut, Quebec or Yukon – see lines 255-259 for the breakdown of responses by province/territory. The lack of data from 3 regions (1 large, 2 small) is likely due to the website and survey only being available in English, the early phase of evaluation, and the lack of outreach into more remote regions in our available time frame for the project. Content was edited to specify the English-only survey was also a study design choice in lines 203-204 and ultimately a limitation in lines 331-332 and lines 355-356.

Comment 6 - Lines 266–267: Consider stating how the toolkit is expected to remain sustainable and up-to-date.

Response 6: A discussion of sustainability is out of the scope of this paper as it depends on future organizational priorities and funding.

Comment 7 - Lines 268–288: Expand on how language barriers or digital literacy may have limited broader engagement.

Response 7: Language is discussed as a limitation in lines 331-332 and lines 355-356. Clarifying information about the number of non-English resources contained in the website is added in lines 328-329. Digital literacy of the website was not assessed nor was digital literacy investigated in the electronic survey, so this aspect is out of scope of this study. It has been added as an area for future study in lines 359-360.

Comment 8 - Lines 135–137: Add clarity about how "impact" is measured — the current phrasing blurs subjective feedback with measurable outcomes.

Response 8: Please see Table 1 for a breakdown of the website impact outcome measures and Supplementary file 1, Appendix C for the survey questions that address these outcome measures.

Comment 9 - Lines 171–173: Clarify how Excel was used for data analysis — were any statistical tests attempted?

Response 9: Content edited in lines 217-218 to specify descriptive statistics.

Comment 10 - Lines 256–258: Reword to reflect that the toolkit "may" support care improvements, as self-reported impact does not confirm behavior change.

Response 10: Please see requested wording change in line 318.

Round 2

Reviewer 2 Report

Comments and Suggestions for Authors

No further comments.